# An Increasing Neutrophil-to-Lymphocyte Ratio Trajectory Predicts Organ Failure in Critically-Ill Male Trauma Patients. An Exploratory Study

**DOI:** 10.3390/healthcare7010042

**Published:** 2019-03-14

**Authors:** Duraid Younan, Joshua Richman, Ahmed Zaky, Jean-Francois Pittet

**Affiliations:** 1Department of Surgery, Division of Acute Care Surgery, The University of Alabama in Birmingham, Birmingham, AL 35249, USA; 2Department of Surgery, University of Alabama in Birmingham, Birmingham, AL 35249, USA; jrichman@uabmc.edu; 3Department of Anesthesiology, University of Alabama at Birmingham, Birmingham, AL 35249, USA; azaky@uabmc.edu (A.Z.); jpittet@uabmc.edu (J.-F.P.)

**Keywords:** trauma, inflammatory cells, outcome

## Abstract

**Background:** Although the association of neutrophil proportions with mortality in trauma patients has recently been shown, there is a paucity of research on the association with other outcomes. We sought to investigate the association of neutrophil proportions with organ failure in critically-ill trauma patients. **Methods:** We reviewed a randomly-selected group of trauma patients admitted to our level-1 trauma intensive care unit between July 2007 and December 2016. Data collected included demographics, injury mechanism and severity (ISS), neutrophil-to-lymphocyte ratio (NLR) at admission and at 24 and 48 hours and organ failure data. NLR patterns during the first 48 hours were divided into two trajectories identified by applying factor and cluster analysis to longitudinal measures. Logistic regression was performed for the association between NLR trajectories and any organ failure; negative binomial regression was used to model the number of organ failures and stage of kidney failure measured by KDIGO classification. **Results:** 207 patients had NLR data at all three time points. The average age was 44.9 years with mean ISS of 20.6. Patients were 72% male and 23% had penetrating trauma. The 74 patients (36%) with Trajectory 1 had a mean NLR at admission of 3.6, which increased to 14.7 at 48 hours. The 133 (64%) patients in Trajectory 2 had a mean NLR at admission of 8.5 which decreased to 6.6 at 48 hours. Mean NLR was different between the two groups at all three time points (all *p* < 0.01). There was no significant difference in ISS, age or gender between the two trajectory groups. Models adjusted for age, gender and ISS showed that relative to those with trajectory 2, patients with the trajectory 1 were more likely to have organ failure OR 2.96 (1.42–6.18; *p* < 0.01), higher number of organ failures IRR 1.50 (1.13–2.00, *p* < 0.01), and degree of AKI IRR 2.06 (1.04–4.06, *p* = 0.04). In all cases, the estimated associations were higher among men vs. women, and all were significant among men, but not women. **Conclusions:** Trauma patients with an increasing NLR trajectory over the first 48 hours had increased risk, number and severity of organ failures. Further research should focus on the mechanisms behind this difference in outcome.

## 1. Introduction

Trauma remains the leading cause of death in the United States among individuals aged 1 to 46 years [1]. Within 48 hours after admission, uncontrolled hemorrhage is responsible for more than fifty percent of trauma-related deaths in both military and civilian hospitals [2]. Survivors of major traumas are often faced with long- and short-term burdens even 1 year after suffering the injury, including problems with mobility (34%), self-care (15%) and activities of daily living (ADL) (51%), pain and discomfort (58%) as well as anxiety and depression (37%) [3].

Trauma and injury result in a systemic body response characterized by acute non-specific immune response that can be associated with weakened immunity and decreased resistance to infection. This can result in damage to multiple organs from the initial cascade of inflammation aggravated by subsequent sepsis to which the body has become vulnerable [4]. Neutrophils are the major organizers of the inflammatory and immunologic response caused by trauma [5].

Recently, inflammatory cell proportions were found to be associated with outcome among cancer patients, Choi et al. [6] found the neutrophil-to-lymphocyte ratio (NLR) to be an independent prognostic factor in patients with gastric cancer, others found a similar association in pancreatic cancer patients [7]; this association with outcome has recently been shown in trauma patients requiring massive transfusions [8]. It is not currently known at this point whether this association is valid outside patients requiring massive transfusions and whether demographic patient factors play a role in this association [9,10]. 

The goal of this study is to explore whether there is an association between NLR in trauma and burn patients at different time points with negative outcomes. We also explore whether this association is influenced by patient demographics. We investigated the clinical association between the proportions of inflammatory cells and organ failure with a view to clinical utilization of this association, and to test the hypothesis that this association would be influenced by patient demographics.

## 2. Materials and Methods

### 2.1. Patient Selection and Variable Definition

After obtaining the Institutional Review Board (IRB) approval, a retrospective analysis of randomly selected trauma patients admitted to our academic level-1 trauma center between July 2007 and December 2016 was conducted. The patients came to the hospital as trauma system activation and required admission per the judgment of the trauma surgeon on duty.

### 2.2. Inclusion and Exclusion Criteria

All adult trauma patients who were admitted to the trauma service over the study period were considered for inclusion; minors (less than 18 years of age), patients with medical diseases that affect the WBC count including leukemia, and patients with human immune deficiency virus (HIV) infection and readmissions were excluded from the study. Data collected included demographics (age, gender, and race), mechanism of injury (blunt, penetrating), injury severity (injury severity score “ISS”), coagulation studies including Prothrombin time (PT), International Normalization Ratio (INR) values and massive transfusion data. We also collected white blood cells count (WBC) including differential count (neutrophils and lymphocytes differential), measures of kidney function (serum creatinine at admission) and hematocrit and platelet levels. The cell proportions are counted by hand (at the hematology laboratory) using 100× magnification in the hematology laboratory. Values were collected upon admission and at 24 and 48 hours. We also collected organ failure data, number of organs failed and the stage of acute renal failure for the first 7 days of admission using KDIGO classification.

### 2.3. Statistical Analysis

Demographics and clinical characteristics were summarized using descriptive statistics (mean and standard deviation for continuous variables; frequency and percentage for categorical variables). Trajectories of NLR were identified by applying factor and cluster analysis to longitudinal measures as implemented in the R package ‘traj’. As an exploratory analysis, Wilcoxon and Chi-square tests were used to evaluate the associations of inflammatory cells with patient and demographic factors and the clinical outcomes of a dichotomous indicator for any organ failure, a count variable of the number of organ failures, and an ordinal variable for degree of kidney failure.

Regression models were used to test for the association between NLR trajectories and outcomes with adjustment for covariates: logistic regression for the binary outcome and negative binomial regression for the number of failures and degree of kidney failure. In addition to models stratified by gender, we also used a regression model that included an interaction term for trajectory and gender. A *p*-value < 0.05 was considered statistically significant in two-tailed statistical tests. All analyses were conducted using R v3.4.2.

## 3. Results

Two hundred and seven patients had NLR data at all three time points. Unadjusted patient characteristics are displayed in Table 1. The average age was 44.9 years with a mean ISS of 20.6. More patient were male than female, and 23% had penetrating trauma. The 74 patients (36%) with increasing trajectory had a mean NLR at admission of 3.6, which increased to 14.7 at 48 hours. The 133 (64%) of patients in decreasing trajectory had a mean NLR at admission of 8.5 which decreased to 6.6 at 48 hours. Mean NLR was different between the two groups at all three time points (all *p* < 0.01). There was no significant difference in ISS, age or gender between the two trajectory groups. When comparing patients in the increasing trajectory by gender (Table 2), there was a significant difference in the NLR48 hours (*p* = 0.05), type of injury (*p* = 0.02) and number of organ failures organ failure (*p* = 0.03).

Models adjusted for age, gender and ISS showed that relative to those with decreasing trajectory, patients with the increasing trajectory were more likely to have organ failure OR 2.96 (1.42–6.18; *p* < 0.01), higher number of organ failures IRR 1.50 (1.13–2.00, *p* < 0.01), and degree of AKI IRR 2.06 (1.04–4.06, *p* = 0.04) (Table 3). In all cases, the estimated associations were higher among men vs. women, and all were significant among men, but not women. Tests for interactions between gender and trajectory had *p*-values of 0.11 (any organ failure) 0.18 (number of organ failures) and 0.22 (KDIGO), i.e., low enough, relative to the modest sample size, to suggest that there may be a genuine interaction.

## 4. Discussion

We found that a trajectory of increasing neutrophil-to-lymphocyte ratio (NLR) over the first 48 hours after admission is associated with the development of organ failure and the number of organs which fail in critically ill male trauma patients; the association was not significant in females. This finding can have important implications in the care of the trauma patients. 

Although neutrophils are described as the first class of infiltrating immune cells at the site of injury, helping to control infection and remove debris [11], traumatic injury also leads to alteration in the function and even the life-span of neutrophils [12]. Dielektasli et al. found NLR to be predictive of mortality in trauma patients admitted to the intensive care unit [13]; this and other studies [8] did not look into the association of NLR with organ failure. Although traditionally immune response has been divided into early and late adaptive responses, with the early one caused by the activation of neutrophils and the late one by the activation of lymphocytes, when the NLR increases, as in the case of patients in trajectory 1, the outcomes change. An early identification of these NLR trajectories can have an impact on the outcome of these patients.

Plenty of research has been done on the effect of sex hormones on outcomes. While after burn injury estrogen is known to have an inhibitory role on the inflammatory response [14] and antecedent ovariectomy was found to improve survival [15], the authors often questioned the presence of a difference in outcome based on gender in trauma literature [9,10]. In this study, we found that an early trajectory of increasing NLR, suggestive of increasing inflammation, is associated with organ failure among male trauma patients but not females. This is the first publication we know of to address gender differences in the association of inflammatory cells and organ failure development.

The patients in the increasing trajectory group were of similar age, and the same gender, race and ISS compared to those in the decreasing trajectory group, but had a higher incidence of organ failure, in addition to a longer ICU, ventilator days and hospital length of stay. While our results are in agreement with other recent publications emphasizing the importance of the association of higher NLR with worse outcomes [8,13], this could represent a maladaptive response of the inflammatory cells in the setting of injury, where the NLR started at a lower value at admission and increased significantly over the following 48 h in this group. This could represent a clinical demonstration of other authors’ reports of the excessive release by neutrophils of elastase, damaging surrounding tissue [16,17], their involvement in tissue damage [18] and infiltration of remote organs contributing to multiple organ failures [19], thus resulting in worse outcomes in these patients. Our study is in line with Dilektasli et al., who demonstrated the prognostic role of NLR at days 2 and 5 in predicting in hospital deaths in trauma patients. Compared to our study, Dilektasli [13] included larger number of patients, did not include ventilator days or ICU lengths of stay as outcomes and included patients with less ISS. Males were more represented in their study, yet they did not specifically explore gender differences.

As inflammation is known to play a role in the pathogenesis of AKI [20,21] and neutrophils are potential surrogate markers of inflammation, certain authors have demonstrated an association between neutrophils and acute kidney injury. Koo et al. and others found that High NLR was associated with postoperative AKI after cardiovascular surgery [22,23]. NLR has also been found to be predictive of AKI in septic patients [24]. Our findings are in agreement with these reports; we found that a trajectory of increasing NLR over the first 48 hours in critically-ill trauma patients is predictive of incidence of AKI, and that it is also predictive of a worse stage of AKI.

The influence of gender differences on inflammatory response has been variably reported in the literature, with some authors attributing it to the acid–base balance of the cellular environment that affects the expression of the X chromosome genes [25], and others to the differing effects of estrogen and testosterone on immune responses, resulting in changes in innate immune cell responses after injury [26]. More recently, gender differences in patient outcomes have also been reported [27,28,29]. Since inflammatory response has been linked to clinical outcomes in critically-ill trauma patients [8], and this differs based on gender, it is conceivable that the outcomes of these patients would differ along gender lines, as well based on the differences in inflammatory cell proportions noted in this study. It is to be mentioned that our observations on gender difference in outcomes is exploratory at the present time, and that further population studies with adequate power are needed to study this association.

Our study suffers from certain limitations. First, it is a retrospective, single-institution study of trauma patients; it is limited by the quality of medical records and the inability to control for all confounders. Second, our data set did not capture other inflammatory markers to help characterize the quality of the inflammatory process. Third, the data are limited to three values (admission, 24 and 48 hours) in the first 48 hours from admission; having more data points during this time period and even a longer follow up period might have provided more detailed information about the trend of these inflammatory cells in the earlier and even later phases, to help better interpret what drives these changes. Fourth, a relatively small number of patients were included in the study. Fifth, more males were represented in our study, and hence, the gender difference observed is exploratory at best.

In conclusion, a trajectory of increasing neutrophil-to-lymphocyte ratio (NLR) over the first 48 hours of admission is associated with the development of organ failure among male trauma patients, and associations may be significantly different by gender; the implications of these findings have not been fully investigated. Further studies are needed to confirm these results.

## Figures and Tables

**Table 1 healthcare-07-00042-t001:** Demographic, injury, and clinical characteristics of trauma patients admitted to the intensive care unit “ICU” in both trajectories.

	Decreasing Trajectory, n = 133	Increasing Trajectory, n = 74	*p*
**DEMOGRAPHICS**			
Mean age (Years)	46 ± 18.3	42.9 ± 16	0.275
Race (%)	-	-	0.110
White	73 (54.9)	29 (39.2)	-
Black	57 (42.8)	43 (58.1)	-
Other	3 (2.3)	2 (2.7)	-
Gender, n (%)	-	-	0.092
Male	93 (69.9)	17 (23)	-
Female	40 (31.1)	57 (77)	-
**INJURY**, n (%)	-	-	0.024
Blunt	111 (83.5)	51 (68.9)	-
Penetrating	22 (16.5)	23 (31.1)	-
**CLINICAL**			
Platelet count at admission (mean, SD)	249 ± 111	235 ± 75.7	0.856
Platelet count at 24 hours (mean, SD)	170.5 ± 107.6	143.2 ± 47.6	0.171
Platelet count at 48 hours (mean, SD)	160.2 ± 116.4	129.8± 45.4	0.092
INR at admission (mean, SD)	1.1 ± 0.2	1.2 ± 0.3	0.145
INR at 24 hours (mean, SD)	1.2 ± 0.2	1.3 ± 0.2	0.037
INR at 48 hours (mean, SD)	1.2 ± 0.2	1.3 ± 0.2	0.002
Hematocrit at admission (mean, SD)	37.6 ± 6.3	37.4 ± 6	0.745
Hematocrit at 24 hours (mean, SD)	30.5 ± 5.9	30.3 ± 5.6	0.709
Hematocrit at 48 hours (mean, SD)	28.4 ± 5.5	28.4 ± 6.3	0.799
Lactic acid on admission (mean, SD)	2.7 ± 2.1	4.8 ± 5	0.000
Lactic acid at 24 hours (mean, SD)	1.4 ± 0.9	1.8 ± 1.3	0.001
Lactic acid at 48 hours (mean, SD)	1.1 ± 0.7	1.3 ± 0.6	0.018
NLR at admission (median, IQR)	8.5 ± 8.7	3.6 ± 3.6	<0.000
NLR at 24 hours (median, IQR)	7 ± 4.7	12.3 ± 10.9	<0.000
NLR at 48 hours (median, IQR)	6.6 ± 3.8	14.7 ± 7.2	<0.000
Organ failure (%)	82 (61.7)	62 (83.8)	0.002
0	51 (38.4)	12 (16.2)	0.001
1	66 (49.6)	37 (50)	-
2	12 (9)	21 (28.4)	-
3	4 (3)	4 (5.4)	-
AKI (KDIGO) (%)	17 (12.8)	25 (33.8)	0.001
AKI stage (KDIGO)	-	-	<0.000
0	116 (87.2)	49 (66.2)	-
1	4 (3)	3 (4)	-
2	2 (1.5)	16 (21.6)	-
3	11 (8.3)	6 (8.1)	-
Massive Transfusion in 24 hours (mean, SD)	123 (92.5)	53 (71.6)	0.0001
ISS (Injury Severity Score)	19.7 ± 10.3	22.1 ± 11.7	0.174
Days on ventilator (mean, SD)	6.6 ± 9.7	12.7 ± 18.1	<0.000
Hospital length of stay (days) (mean, SD)	15.8 ± 13	23.1 ± 19.3	0.002
ICU length of stay (days) (mean, SD)	10.4 ± 10.6	17.2 ± 19.8	<0.000
Dead, n (%)	7 (5.3)	3 (4)	0.752

**Table 2 healthcare-07-00042-t002:** Demographic, injury, and clinical characteristics of trauma patients in increasing trajectory (1) in both gender.

	Females n = 17	Males n = 57	*p*
**DEMOGRAPHICS**			
Mean age (Years)	49.2 ± 17.3	41 ± 15.3	0.275
Race (%)	-	-	0.110
White	12	17	-
Black	5	38	-
Other	0	2	-
**INJURY**, n (%)	-	-	0.024
Blunt	17	34	-
Penetrating	0 (0)	23 (0)	-
**CLINICAL**			
Platelet count at admission (mean, SD)	245 ± 68.5	232 ± 78.1	0.422
Platelet count at 24 hours (mean, SD)	144.8 ± 59.3	142.7 ± 44.1	0.837
Platelet count at 48 hours (mean, SD)	118.5 ± 46.6	133.2 ± 44.9	0.199
INR at admission (mean, SD)	1.17 ± 0.26	1.18 ± 0.27	0.700
INR at 24 hours (mean, SD)	1.26 ± 0.19	-	0
INR at 48 hours (mean, SD)	1.32 ± 0.28	1.27 ± 0.13	0.638
Hematocrit at admission (mean, SD)	34.5 ± 4.3	38.3 ± 6.3	0.004
Lactic acid on admission (mean, SD)	3.2 ± 2.8	5.3 ± 5.5	0.046
Lactic acid at 24 hours (mean, SD)	1.44 ± 0.64	1.94 ± 1.37	0.093
Lactic acid at 48 hours (mean, SD)	1.05 ± 0.34	1.3 ± 0.65	0.161
NLR at admission (median, IQR)	3.17 ± 1.65	3.67 ± 3.98	0.24
NLR at 24 hours (median, IQR)	9.8 ± 4.15	12.98 ± 12.16	0.393
NLR at 48 hours (median, IQR)	11.63 ± 3.78	15.65 ± 7.75	0.046
Organ failure (%)	11	51	0.025
0	6	6	0.067
1	8	29	-
2	3	18	-
3	0	4	-
AKI (KDIGO) (%)	3	22	0.152
AKI stage (KDIGO)	-	-	0.106
0	14	35	-
1	1	2	-
2	0	16	-
3	2	4	-
Massive Transfusion in 24 hours (mean, SD)	4 (23)	17 (29)	0.751
ISS (Injury Severity Score) (mean, SD)	20.9 ± 11.5	22.5 ± 11.8	0.563
Days on ventilator (mean, SD)	9.4 ± 9.7	13.6 ± 19.9	0.394
Hospital length of stay (days) (mean, SD)	19.5 ± 10.3	24.2 ± 21.3	0.852
ICU length of stay (days) (mean, SD)	13.3 ± 9.7	18.3 ± 21.8	0.743
Dead, n (%)	0	3	0.581

**Table 3 healthcare-07-00042-t003:** Multivariable regression analysis for the association between NLR in “increasing’ trajectory and outcomes compared to those in “decreasing” trajectory among trauma patients admitted to the intensive care unit.

Trauma Patients in “Increasing” Trajectory Group*,* N = 74 vs. “Decreasing” Trajectory	Adjusted Result	Confidence Limits	**p*	*p for Interaction*
**Organ failure (OR)**				0.11
Overall:	2.96	1.42–6.18	<0.01	
Men	4.68	1.77–12.37	<0.01	
Women	1.25	0.37–4.25	0.72	
**Number of organs failed (IRR)**				0.18
Overall:	1.5	1.13–2.00	<0.01	
Men	1.69	1.22–2.35	<0.01	
Women	1.01	0.53–1.90	0.95	
**AKI stage (KDIGO) (IRR)**				0.22
Overall:	2.06	1.04–4.06	0.04	
Men	2.47	1.17–5.25	0.02	
Women:	1.01	0.23–4.39	0.99	

* Each row in the table represents a separate multivariable negative binomial regression model, adjusted for age, sex, injury type and injury severity score.

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
