# Peer review of "An Increasing Neutrophil-to-Lymphocyte Ratio Trajectory Predicts Organ Failure in Critically-Ill Male Trauma Patients. An Exploratory Study"

_healthcare, 2019, doi:10.3390/healthcare7010042_

Round 1

Reviewer 1 Report

This observational study has focused on an interesting topic and the collection of the data has been properly described. However, I am not convinced regarding the gender difference, since the head injury pattern i different in men an women. In order to convince me and other reader of important gender differences. A subgroup of patients with blunt injury should be compared regarding the presence of gender difference. 

A separate analysis of blunt and penetrating injury may also generate potentially important hypothesis. 

Presenting data as curves instead of data points and boxes could help to grasp the pattern better. In addition the figure legend should explain the graph, percentile range or 95 % CI?   

Author Response

Reviewer1:

    This observational study has focused on an interesting topic and the collection of the data has been properly described. However, I am not convinced regarding the gender difference, since the head injury pattern i different in men an women. In order to convince me and other reader of important gender differences. A subgroup of patients with blunt injury should be compared regarding the presence of gender difference. 

    A separate analysis of blunt and penetrating injury may also generate potentially important hypothesis. 

    Presenting data as curves instead of data points and boxes could help to grasp the pattern better. In addition the figure legend should explain the graph, percentile range or 95 % CI? 

Authors’ response:

We thank the reviewer for his comment. The primary goal of this study was to explore whether elevated NLR is associated with outcomes. The secondary goal was to study whether this association is influenced by demographic patient factors. The observation that male patients are associated with negative and more significant outcomes is hypothesis generating at this point. We have emphasized this meaning (lines 54-61,85-88, and 159-161). As is known, exploring effects of gender would need a large number of patients.

    We have added a figure legend to explain the figure (Lines 263-266).

Reviewer 2 Report

Neutrophil-to lymphocyte ratio, NLR, which reflects inflammatory and immunologic response is recently reported as a prognostic factor of cancer, and its validity in trauma is under investigation. The authors claim an association between NLRs and long-term outcome of trauma by statistical analysis of patient. A group of patient of increasing trend of NLR within 48h after admission are significantly more susceptible to organ failure than anther group of higher initial value and decreasing trend. This association was only prominent in male group. This is a well organized clinical data, but still can be improved.

Major

1. The possible contribution of medication to the decreasing trend, as well as to the better outcome, need to be investigated. 

2. Physiological parameter, including body temperature, blood pressure etc at admission also need to be investigated in the association with initial NLR.

3. The result part need to be more in detail. And values should be described as means +/- SE or STD

Minor

1. 90 “……"

2. 99 “.."

3. Table 2 lacks values in some (). 

Author Response

Reviewer 2:

    Neutrophil-to lymphocyte ratio, NLR, which reflects inflammatory and immunologic response is recently reported as a prognostic factor of cancer, and its validity in trauma is under investigation. The authors claim an association between NLRs and long-term outcome of trauma by statistical analysis of patient. A group of patient of increasing trend of NLR within 48h after admission are significantly more susceptible to organ failure than another group of higher initial value and decreasing trend. This association was only prominent in male group. This is a well organized clinical data, but still can be improved.

    Major

1.  The possible contribution of medication to the decreasing trend, as well as to the better outcome, need to be investigated.

 Authors response:

These patients were relatively healthy trauma patients who were not suffering from multiple co-morbidities. All patients received the standard of care for patients based on their severity and hospital course. In our models of logistic regression we adjusted for factors that were more likely to influence the association between NLR and outcomes.

2.  Physiological parameter, including body temperature, blood pressure etc at admission also need to be investigated in the association with initial NLR.

Authors response:

We agree with the reviewer on his comments. However, the goal of this study is a proof of concept at this time, given the retrospective and observational nature of the study. We believe that the details of demographics and vital signs and their influence of outcomes better suit a large adequately powered study.

    3. The result part need to be more in detail. And values should be described as means +/- SE or STD

Authors response:

Changes made

Round 2

Reviewer 1 Report

The manuscript has improved, and  I suggest that the title should be 

An Increasing Neutrophil-to-lymphocyte Ratio Trajectory Predicts Organ Failure In Critically-ill  Male Trauma Patients but Not in Females. An 5 Exploratory Study

The addition an exploratory study is not needed in the title.

The outcome in the female group was underpowered and having the suggested title may be missleading. 

an alternative title that may be more acceptable could be: An Increasing Neutrophil-to-lymphocyte Ratio Trajectory Predicts Organ  Failure In Critically-ill Male Trauma Patients but maybe not in Females

Author Response

An Increasing Neutrophil-to-lymphocyte Ratio Trajectory Predicts Organ Failure In Critically-ill  Male Trauma Patients but Not in Females. An  Exploratory Study

Authors' response:

We very much thank the reviewer for his comments and we totally agree with them.

We have changed the title of the manuscript to the new suggested title.

Please accept our deepest appreciation.